# The Contribution of Individual Characteristics of Managers to the Success of Equivalency Education Programs of the Community Learning Center in Indonesia

**Sucipto [1,\*], Bambang Budi Wiyono [2,\*], Ach. Rasyad [1], Umi Dayati [1] and Lasi Purwito [1]**

1   NonFormal Education Department, Faculty of Education, Universitas Negeri Malang, Malang 65145, Indonesia; ach.rasyad.fip@um.ac.id (A.R.); umi.dayati.fip@um.ac.id (U.D.); lasi.purwito.fip@um.ac.id (L.P.)
2   Educational Administration Department, Faculty of Education, Universitas Negeri Malang, Malang 65145, Indonesia
\*   Correspondence: sucipto.fip@um.ac.id (S.); bambang.budi.fip@um.ac.id (B.B.W.)

**Abstract:** The community learning center provides a means of completing compulsory education and increasing the level of educational participation in Indonesia. Through this program, regardless of age, people can continue their education to a level equivalent to junior or senior high school. However, factors that contribute to the success of the program are still in question. This study aimed to determine the contribution of individual characteristics, including work experience, work perception, work motivation, and work discipline, to the success of the equivalency education program at the community learning center. This research was conducted in East Java using a quantitative approach and an explanatory research design. The research sample was taken from 101 program managers through random sampling techniques and the Slovin formula. The data were collected using a questionnaire and then analyzed using multiple regression techniques. The results showed that the education program had high success, with the participants achieving examination pass rates of 96.4% in package B, which is equivalent to junior high school, and 98.2% in package C, which is equivalent to senior high school. The average involvement of community members in the program implementation was 68.6%, and the financial support reached 75.5%. The work experience, work perception, work motivation, and work discipline of the program managers had a significant effect, both simultaneously and partially, on the success of the equivalency education program. In addition, the four independent variables also had a significant predictive effect on the dependent variable, with a simultaneous contribution of 53.8%. Based on these findings, it was recommended that these factors be considered to improve the success, sustainability, and development of education programs at the community learning center.

**Keywords:** equivalency education program; community learning center; work experience; work perception; work motivation; work discipline

## 1. Introduction

The community learning center (CLC), which operates outside of the formal education system in both urban and rural areas in several Asian countries, especially in Indonesia, was assessed by UNESCO to be running smoothly, meeting targets, and successfully achieving its goals. A CLC is considered capable of playing a role in driving the literacy movement, increasing the ability of a community to participate more broadly in development, and to adapt to societal changes and progress, and improve the standard of living of a community [1]. Furthermore, the program for accelerating the eradication of illiteracy, the study groups of package B (equivalent to junior high school) and package C (equivalent to senior high school), a life skills course, a training program, a community empowerment program, and an early childhood education program provide a meaningful contribution to

the success of the government towards the goal of lifelong education and education for all. All of the programs are managed by the CLC. Several studies showed similar results, although the context and substance differ, including: in the Indonesian territory [2], West Java [3], Central Java [4], Central Kalimantan [5], and Malang, East Java [6]. Therefore, to improve the quality and equity of education, it is necessary to increase the success of the CLC equivalency program. The characteristics of the managers are thought to be the main variables that influence the success of the program. Based on this rationale, the research was conducted to determine the effect of manager characteristics on the success of the CLS equivalency program.

## 1.1. Literature Review

The community learning center (CLC) was developed from, by, and for society [7]. This principle means that the CLC is in the midst of people's lives and not only aspires to meet the learning needs of the surrounding community, but also uses high local potential as a basis for developing its service programs. The commitment and responsibility of the surrounding community also has to be cultivated to ensure the sustainability of the CLC itself. Therefore, the development of the CLC needs to maximize the engagement of the surrounding community so that all levels of society are proportionally involved in every program management process, from decision making and planning to implementation, monitoring, and evaluation [8].

The presence of the CLC in people's lives is important in both developed and developing countries, including Indonesia. Education program services and community empowerment can accelerate efforts to improve the quality of the human resources needed for further development. These services have real-world benefits to the community [9]. For example, the literacy education program and the study group programs of package A (equivalent to elementary school), package B (equivalent to junior high school), and package C (equivalent to high school), apart from being considered expansive and progressive, have succeeded in increasing the literacy index, the total participation rate of the directorate of basic education, and the total participation rate of the directorate of secondary education. These programs are also considered successful in increasing the skills and participation of society in development [10,11].

The success of the CLC program, especially the equivalency program, cannot be separated from the role and performance of CLC managers. The success or failure of the CLC equivalency program is largely determined by the characteristics of CLC managers, including those responsible for the equivalency program. The managers of the CLC equivalency program should be able to: (1) identify the learning needs of society, (2) prepare plans, (3) guide learning, (4) assess learning outcomes, and (5) report the implementation and the results of activities to relevant officials [7].

The implementation of learning programs will succeed in achieving the targeted goals and objectives, and in providing functional benefits to participants, if the CLC program managers have adequate managerial and professional competencies. The qualifications and competencies of educators and administrators of the early childhood education program in the PNF unit, including the CLC in almost all regions in Indonesia, were still inadequate [2]. This is one of the reasons that the implementation of the early childhood education program in the PNF unit is difficult and tends to be formal; the learning output, in the form of behavioral change, especially in the cognitive and affective domains, is less than optimal, and the social and economic impacts were less significant than expected.

Organizational performance cannot be separated from the performance of individuals in the organization, as employee performance determines the success of company activities. The company will be able to survive in a competitive environment if it is supported by employees who are competent and work well in their fields. Employee performance is influenced by motivation, ability, knowledge, expertise, education, experience, training, interests, attitudes, and personality, as well as physical, physiological, social, and self-esteem needs [12]. Empirical research results show that employees who have adequate

and relevant education and experience that support their skills will find it easier to achieve high performance in carrying out their work responsibilities and routine tasks. Employee abilities are strongly influenced by education and work experience [13]. The results of research on SMEs in Bangkalan show that one's work experience is important for improving the quality and success of the work involved because continuously repeating tasks over time can strengthen the accuracy, speed, and quality of the work itself [14].

Moreover, from the perspective of organizational behavior studies, the performance, competence, and ability displayed by a person in carrying out their duties and functions in an organization, whether business, public, or social organizations, is primarily determined by the human element, followed by the financial element, including materials, and finally, the technology used for the development and progress of the organization itself. A number of studies indicated that the quality of human resources is linearly related to the quality of performance displayed by individuals [15]. This reality is understandable because the main actors of the organization are humans, while money, goods, and technologies are the targets and tools that can support the smooth and successful performance of human beings [16]. If human resources do not have adequate skills and sufficient work experience, then the smooth and successful implementation of individual tasks will be compromised, regardless of the amount of funding, the intricacy of the infrastructure, and the sophistication of the technology.

Everyone in an organization has individual characteristics that are unique. Some characteristics are inherently static and relatively permanent, for example, the basic innate features that contribute to an individual's potential, including physical characteristics, intelligence, talents, and interests. Conversely, other characteristics develop dynamically and progressively as the individual journeys through life, continuing to strengthen and quantitatively grow, including, for example, educational, work, and organizational experience, among other types of experience. Some aspects develop dynamically and fluctuate, appearing suddenly and unpredictably, such as perceptions, motivations, emotions, efficacy, resilience, discipline, and commitment [12]. These individual aspects are normally considered psychological aspects, which in addition to being dynamic and fluctuating, are also regarded as mysterious, latent, and hidden: they can be captured and interpreted only by their outward manifestations. These individual characteristics are very important to thoroughly understand in detail, so they can be used as assessment criteria for every leader involved in decision making and various organizational interests. One of the goals for the organization and its management, apart from making tasks appropriate and enjoyable for the person performing them, is to make it easier to fill positions after promotions, job transfers, and other staff changes.

Most research findings related to the impact of individual aspects on a person's abilities, competencies, performance, and/or achievements in various types of organizations reveal linear and significant associations, although the level of linearity and degree of significance vary. Static individual characteristics, which include gender, age, intelligence, talents, and interests, are often used as predictors of the performance of employees, managers, or directors in various organizations or companies. Similarly, dynamic individual aspects, such as formal education, non-formal education, work, and organizational experience, tend to be treated as predictor variables that affect the ability, competence, performance, achievement, and success of leaders and employees of various organizations. Likewise, fluctuating individual aspects, which are also called psychological aspects, including perception, motivation, connection, emotion, efficacy, resilience, discipline, and commitment, are often analyzed as independent variables that affect the ability, competence, performance, achievement, and success of employees, managers, or directors in business, public, and social organizations [17]. However, these studies showed varied results, indicating that further research is required.

Research in West Java showed that, among the ability, motivation, and commitment of employees, motivation had the greatest influence on employee performance. Competence was the second most influential, and commitment had the smallest effect on employee

performance. Taken together, ability, motivation, and commitment explained 70.03% of employee performance. Other factors outside the scope of research also affected employee performance, contributing 29.97%; such other variables include work environment, compensation, organizational culture, communication, and leadership [18].

Research findings at several regional hospitals in Central Java showed that work discipline, work motivation, job satisfaction, and work competence had a significant effect on employee work commitment. The magnitude of the effect on employee commitment was ranked based on the regression coefficient. Work motivation had the highest ranking, with a coefficient of 1.07, followed by job satisfaction with 0.78, work competence with 0.73, and, finally, work discipline with 0.50 [19].

Moreover, in a study on local water company employees in Malang Raya, path analysis showed that work motivation directly affected employee performance and indirectly influenced it through organizational commitment to employee performance; all empirical test results were high [20]. The overall results for the direct or indirect effect of work discipline on employee performance were lower. The motivation to commitment was 0.477, motivation to performance was 0.218, and motivation to performance through commitment was 0.758. Meanwhile, discipline towards performance was 0.159, discipline towards commitment was 0.186, and discipline towards performance through commitment was 0.370 [20].

On the other hand, the results of several studies conducted at formal educational organizations showed varied results. Sofian et al.'s (2019) research showed that work discipline had a significant effect on teacher work performance, with a regression coefficient of 0.680 [21]. Citriadin et al. (2019) showed that attitude affected teacher performance with a regression coefficient of 0.467 [22]. Rahayu's research (2019) showed that intelligence had a significant influence on the professional competence of teachers, with a coefficient of 0.146 [23]. The results of Alwi's research (2021) confirmed that the organizational citizenship behavior of teachers was affected by emotional intelligence (with a coefficient of 0.166), the big five personality traits (0.310), attitudes, including organizational commitment (0.280), and perceptions of organizational justice (0.214) [24].

Based on some of the results of these studies, it can be concluded that individual characteristics affect the performance or contribute to the success of leaders, but they still need to be investigated in greater depth, especially in CLCs. The CLC is a social organization engaged in education and community empowerment as a non-formal education unit. The number of facilities is relatively large and continues to grow every year, and they are distributed throughout all regions in Indonesia. However, only about 10% were accredited. Programs facilitated by CLCs have greatly contributed not only to an increase in literacy skills, allowing participants to keep up with evolving information and improve their life skills to obtain decent jobs and income, but also to improvements in community participation in development, adaptability to the progress of civilization, and the quality of life of the community.

The CLC program is generally quite responsive to the diverse learning needs of the community. The main programs are basic and functional literacy packages, as well as equivalency and skills training packages, and are quantitatively achieving targets set out in the contract. However, community empowerment programs, community reading parks, and partnership programs generally do not achieved optimal results. Qualitatively, the socio-economic impact and benefits are not optimized for improving the quality of life of society. This is partly attributable to the individual aspects of the CLC program manager, who is responsible for the effectiveness and success of the CLC program. To strengthen the CLC program, research is needed to determine how the personal characteristics of the leadership influences its success.

Success can be simply defined as achievement, implementation, and performance. The success of the CLC equivalency program thus refers to its implementation, achievement, and performance. This research focused on two aspects of the equivalency program to evaluate its success, namely, the output/outcome aspect and the community/stakeholder

support aspect. The indicator of outcomes was the percentage of graduates per year, and the indicator of community/stakeholder support was the percentage of community involvement and funds that were collected and donated by the community for the implementation of the equivalency program. The success of the CLC equivalency program, apart from being an indicator of its performance as an organization, was also interpreted as an indicator of the success of the equivalency program manager as an individual who is responsible for managing it.

From the perspective of organizational behavior theory, the management of literacy programs is the main element in the CLC organization, in addition to funds, materials, and technology. However, the existing conditions are somewhat ironic; realizing the importance of human beings as the core of the organization, the prerequisites of a CLC manager were made very basic, apparently open to almost anyone, without strict qualifications. In the CLC organizational structure, the program manager is referred to as the coordinator of the program unit. The manager of CLC has five main tasks, namely, identifying community learning needs, planning program implementation according to community needs, fostering program implementation, evaluating ongoing programs, and reporting the program results to the relevant officials. A CLC program manager must be able to accommodate all programs in the community and ensure that it runs effectively. The learning programs include functional literacy programs, equivalency programs, business study groups, development of internship programs, and skills courses.

Individual aspects, including static, dynamic, and fluctuating characteristics, are interesting to study as predictor variables on the performance of program managers, as well as on the success of the CLC equivalency program. Each characteristic of the individual aspect was selected based on the empirical results of previous research. Analytical-inferential studies on the individual aspects of social organization managers in CLC are still scarce to almost non-existent. Several study themes related to existing CLCs tend to be in the form of exploratory descriptive and narrative descriptive analyses. The sites and loci of analytical-inferential studies on the individual aspects of program managers were also conducted almost entirely in business and public organizations. Therefore, it was necessary to conduct this research.

*1.2. Research Hypotheses*

This study aimed to: (1) describe the success of the CLC equivalency program, particularly aspects of outcomes/outputs and aspects of community support; (2) describe work experience, work perception, work motivation, and work discipline in managing the equivalency program at a CLC; and (3) test the partial and simultaneous effects of work experience, work perception, work motivation, and work discipline on the success of the CLC equivalency program. The hypothesis tested in this study addresses the question of whether work experience, work perception, work motivation, and work discipline simultaneously and/or partially affect the success of the equivalency education program in CLCs. The coefficient of the contribution of each variable varies, and this was studied in depth in this research. Through this research, useful information was obtained to maintain and improve the quality of education at CLCs. Theoretically, the results of this study can also be used to develop a theory of human resource management in non-formal educational institutions.

Based on the present research objectives, and previous theory and empirical results, the major hypothesis that was tested in this study was that work experience, work perception, work motivation, and work discipline simultaneously significantly influence the success of the CLC equivalency programs. The major hypothesis was broken down into four minor hypotheses, namely: (1) the manager's work experience has a positive effect on the success of the equivalency program, (2) the manager's work perception has a positive effect on the success of the equivalency program, (3) the manager's work motivation has a positive effect on the success of the equivalency program, and (4) the manager's work discipline has a positive effect on the success of the equivalency program.

## 2. Materials and Methods

### 2.1. Research Design

This study used a quantitative approach designed to study cause/effect correlation. The research was conducted from 2019 to 2020. The process was based on deductive-inductive logic, starting from formulating the problem, deepening the theoretical framework, formulating hypotheses, and then verifying empirical support using statistics [25]. From the perspective of the philosophy of science, this research approach was based on the positivist paradigm, in which the truth-seeking process must follow scientific principles, namely, logical, objective, empirical, systematic, and measured. In this study, the success of the CLC program, which is used as the criterion variable (Y), was limited to the equivalency program, which included the study group programs of package B (equivalent to junior high school) and package C (equivalent to senior high school); the CLC organizes these programs.

Several individual aspects are thought to affect the success of the CLC equivalency program, namely, work experience, work perception, work motivation, and work discipline, which were theoretically and empirically tested in research in business and public organizations. The individual aspects were positioned as predictor variables (X1, X2, X3, and X4). The analytical method used to review various possible relationships between variables in this study was based on organizational behavior theory in accordance with McShane (2018), Robbins (2017), and George (2012), and was supplemented with some expert opinions and relevant research findings [12,15,16].

Following the formulation of the problem and the research objectives in this study, the five research variables were divided into two groups depending on their position and the direction of the relationship, namely: (1) independent or predictor variables, and (2) dependent or criterion variables. Predictor variables that were considered to affect the independent variable included: (1) work experience, (2) work perception, (3) work motivation, and (4) work discipline. In the context of organizational behavior science, these four variables are referred to as individual aspects [12]. The dependent variable or criterion variable in this study was the success of the CLC equivalency program.

The schematic design regarding the position and direction of the relationship between the independent and the dependent variables, both partially and simultaneously, is described in Figure 1 as follows.

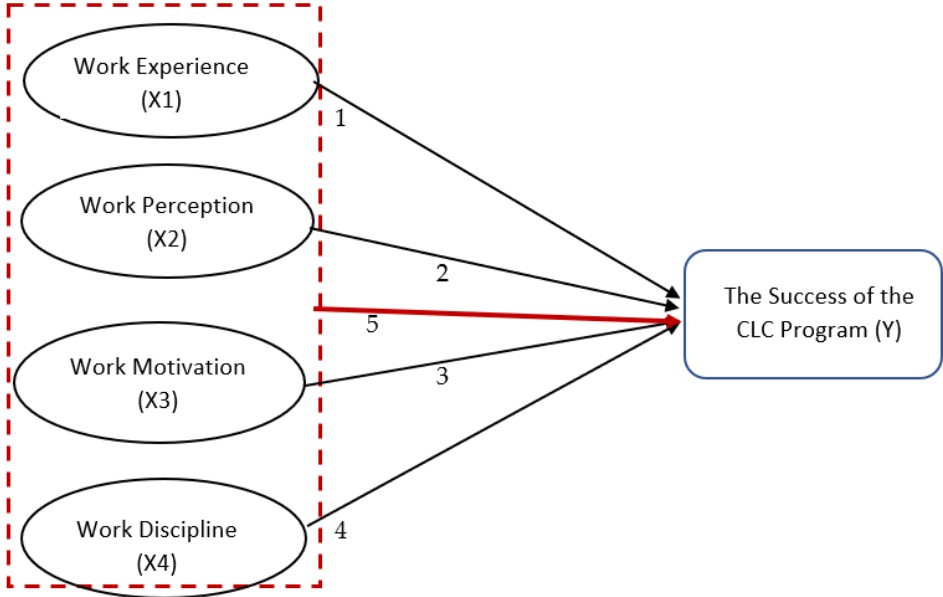

**Figure 1.** The relationship between independent and dependent variables.

### 2.2. Participants

The population in this study comprised of the managers of the CLC equivalency program in Malang Regency and Malang City. Together, Malang Regency and Malang City have 60 CLC units: 45 CLCs in Malang Regency and 15 CLCs in Malang City. Each CLC has two program managers. Thus, the total population in this study was 120 people in charge of managing the equivalency education program. Sampling was based on the Slovin formula with an error of 0.5%, and the number 92.30 was rounded to 92. Thus, using the proportional random sampling technique, the number of samples was 92 people who managed the study group programs of package B, which is equivalent to junior high school, and package C, which is equivalent to senior high school. To increase the representation, the research sample was increased by 10% to 101 managers. The characteristics of the research sample in terms of gender, educational background, age range, and work experience as managers are presented in Tables 1–4.

**Table 1.** The distribution of the research sample based on gender.

| No. | Gender | Frequency | Percent |
|-----|--------|-----------|---------|
| 1 | Male | 61 | 60.4% |
| 2 | Female | 40 | 39.6% |

**Table 2.** The distribution of the research sample based on educational background.

| No. | Education Level | Frequency | Percent |
|-----|-----------------|-----------|---------|
| 1 | Senior High School | 14 | 13.9% |
| 2 | Diploma III | 3 | 3.0% |
| 3 | Bachelor | 70 | 69.3% |
| 4 | Master | 14 | 13.9% |

**Table 3.** The distribution of the research sample based on work experience.

| No. | Work Duration | Frequency | Percent |
|-----|---------------|-----------|---------|
| 1 | 1–5 years | 23 | 22.8% |
| 2 | 6–10 years | 53 | 52.5% |
| 3 | 11–15 years | 19 | 18.8% |
| 4 | 16–20 years | 6 | 5.9% |

**Table 4.** The distribution of the research sample based on age range.

| No. | Age Range | Frequency | Percent |
|-----|-----------|-----------|---------|
| 1 | 21–30 years | 29 | 28.7% |
| 2 | 31–40 years | 36 | 35.6% |
| 3 | 41–50 years | 25 | 24.8% |
| 4 | 51–60 years | 11 | 10.9% |

Tables 1–4 demonstrate that the managers in the sample varied in terms of gender, educational background, work experience, and age range. Most had an undergraduate education, the ages ranged from 31 to 40 years, and they had a working period of 6–10 years as a CLC manager. The characteristics of the sample reflected those of the population. Thus, the sample of this study was representative.

Each CLC has tutors, the number of which depends on the size of the institution, with a minimum of five, a maximum of 17, and an average of nine tutors. The number of staff in each institution is 1–5 personnel, with an average of three personnel. The number of students varies with the size of the institution, with a minimum of 10, a maximum of 60, and an average of 23. Thus, among all 101 institutions, there were 2290 participants, 855 tutors,

and 228 staff. The students follow the learning program according to the package that they are enrolled in. The prerequisites of package B are completion of elementary school, package A, or other community education that is equivalent to completion of elementary school. In addition, the prerequisites of package C include completion of junior high school, package A, or other community education that is equivalent to completion of junior high school. The learning program is scheduled and carried out flexibly by emphasizing the use of adult learning strategies. In the implementation of education, each CLC is led by the head of the organization and assisted by staff such as treasurers, secretaries, and coordinators or program managers, depending on the organizational structure of each institution. Program managers are responsible for identifying community learning needs, planning programs, fostering and evaluating program implementation, and reporting the results of program implementation to the relevant officials. The program manager is also assisted by staff in conducting their duties so that education at the CLCs can be delivered effectively.

### 2.3. Research Instruments

Based on the target characteristics of the study, data were collected using a questionnaire. The questionnaire instrument was developed based on the construct variables studied, namely, predictor variables, and contained questions or statements related to individual aspects, which included work experience, work perception, work motivation, and work discipline. The questionnaire was adapted from instruments developed previously [12,15,16]. The questionnaire forms included closed and open questions. The work experience instrument consisted of five items that referred to work experience as a valuable lesson, increasing knowledge, improving skills, improving performance, and promoting job completion. The work perception instrument consisted of seven items, namely, whether the CLS education package is perceived as beneficial to the community, increasing participation of society in development, advancing society, noble work that is fun and light, and in devotion to God. The work motivation instrument consisted of five items, namely, motivation to earn a salary, obtain health insurance, get a promotion, be in a leadership role, and cooperate with colleagues. The work discipline instrument consisted of six items, namely, arriving on time, leaving on time, obeying regulations, avoiding prohibitions, accomplishing achievements, and being enthusiastic about achieving program targets. Each item had seven answers: "Strongly Agree" = 7, "Agree" = 6, "Slightly Agree" = 5, "Somewhat Agree" = 4, "Slightly Disagree" = 3, "Disagree" = 2, and "Strongly Disagree" = 1.

The instrument for evaluating the success of the CLC equivalency program (the criterion variable) was developed based on the construct variables and guidelines used in the field, namely, the CLC standard book set by the Directorate General of Early Education of the Ministry of Education and Culture in 2012 and the Directorate of Public Education Department in 2014. This instrument consists of seven items: the number of students, number of graduates, average grades achieved by students, student attendance, tutor attendance, community involvement in education, and community funding support.

To ensure validity and reliability, the instrument was tested on the equivalency program managers at CLCs in Pasuruan Regency and city. The results of the analysis of the test instrument were $r_{ii} = 0.977$ for work experience, $r_{ii} = 0.981$ for work perception, $r_{ii} = 0.978$ for work motivation, $r_{ii} = 0.984$ for work discipline, and $r_{ii} = 0.984$ for CLC success. The results of the instrument analysis showed a validity coefficient for each item of >0.3 and Cronbach's Alpha reliability coefficient of >0.7. Thus, the research instrument showed good validity and reliability.

### 2.4. Data Analysis

The collected data were analyzed quantitatively through descriptive and inferential statistics. Descriptive statistics were used to address the research objectives of points one and two, specifically regarding the distribution and trends of each individual aspect of

the CLC program manager. The inferential statistical analysis technique, namely, multiple regression, was used to address research item three: determining the effects of the independent variables on the dependent variable, both partially and simultaneously. In addition, before testing the hypothesis, the classical assumption test was also carried out, which included tests for normality, linearity, multicollinearity, autocorrelation, and heteroscedasticity. The analysis process was implemented in the SPSS statistical analysis program package for Windows [26–29].

## 3. Results

### 3.1. The Success of the CLC Equivalency Program

The success of the equivalency education program in this study was measured by the results of the equivalency exam and analyzed based on two aspects: (1) the output/results achieved in the form of the percentage of participants passing the exam of package B, or package C, and (2) community support in the form of the percentage of members of society involved in the implementation of the equivalency study group package and the percentage of public funds that can be collected for the implementation of the equivalency study group package.

Participants of package B passed the exam at a rate of 96.4%, which was classified as very high. Furthermore, the participants of package C passed the exam at a rate of 98.2%, which was also considered very high. Based on the output/graduates of packages B and C equivalency programs, the average success rate of CLCs in Malang Regency and city is 97.3%, and thus classified as very high.

Public support for the implementation of the CLC equivalency program was generally high. For the study group of package B, 82% of respondents stated that 67% of the community members outside of the relatives of the daily CLC management were involved in program implementation, which is high. For the study group of package C, 78% of respondents stated that 70.2% of society members outside of the CLC management were involved in program implementation, which is also high. On average, 80% of respondents stated that an average of 68.6% of the community members were involved in the implementation of the CLC equivalency program, which, too, is considered high.

Public funds that are raised to support the program were generally high. For package B, 81% of respondents stated that the public funds that could be raised to support program implementation reached 65%, which is high. For the study group of package C, 91% of respondents stated that the public funds that could be raised to support program implementation reached 86%, which is classified as very high.

The results of the descriptive data analysis of each research variable were grouped into five categories: very good, good, sufficient, poor, and very poor. These are summarized in Table 5. The classification of the categories was obtained by subtracting the minimum score from the maximum score, divided by the number of classes.

**Table 5.** The results of the descriptive analysis of independent and dependent variables.

| No. | Variable | Mean | Standard Deviation | Score Category |
|-----|----------|------|--------------------|----------------|
| 1 | Work Experience | 24.74 | 7.803 | Good |
| 2 | Work Perception | 34.50 | 9.869 | Good |
| 3 | Motivation | 25.89 | 7.377 | Good |
| 4 | Work Discipline | 30.96 | 9.245 | Good |
| 5 | Success of the CLC Program | 25.49 | 8.925 | Good |

From the results in Table 5, it can be concluded that the average levels of work experience, perception, motivation, and discipline could be categorized as "good", with relatively small standard deviations. Likewise, the average success of CLC was also categorized as good, with a relatively small standard deviation.

### 3.2. The Influence of the Work Experience, Work Perception, Motivation, and Work Discipline of Managers on the Success of the CLC Equivalency Program

The next step was hypothesis testing, namely, testing the effect of the independent variables on the dependent variable, either simultaneously or partially. Following the research design, regression analysis was used to test the hypothesis. Before the analysis, assumptions were tested. The results of the classical assumption test showed that most of the data met the analysis requirements. The data were linear and did not show multi-collinearity, autocorrelation, or heteroscedasticity. For this reason, regression analysis was carried out to test the hypothesis. With a total sample size of 101, the results of the overall regression analysis are presented in Table 6.

**Table 6.** Summary of regression model analysis results.

| Model | R | R-Squared | Adjusted R-Squared | Std. Error of the Estimate | Change Statistics | | | | |
|---|---|---|---|---|---|---|---|---|---|
| | | | | | R-Squared Change | F Change | df1 | df2 | Sig. F Change |
| 1 | 0.734 [a] | 0.538 | 0.519 | 6.189 | 0.538 | 27.981 | 4 | 96 | 0.000 |

| ANOVA [b] | | | | | | |
|---|---|---|---|---|---|---|
| Model | | | Sum of Squares | df | Mean Square | F | Sig. |
| 1 | Regression | | 4240.774 | 4 | 1060.194 | 27.327 | 0.000 [a] |
| | Residual | | 3724.454 | 96 | 38.796 | | |
| | Total | | 7965.228 | 100 | | | |

[a] Predictors: (constant), work discipline, work experience, work perception, work motivation. [b] Dependent variable: success of the equivalency program.

Looking at Table 6, the overall R-value was 0.734, and the R-Squared value was 0.530, with an F value of 27.981 and a Sig. F-value of <0.01. The error probability value was below 0.05 or even 0.01. Thus, it can be concluded that the null hypothesis (H0) was rejected, and the alternative hypothesis (H1) was accepted; that is, work experience, work perception, motivation, and work discipline have a simultaneous, significant positive effect on the success of the CLC equivalency education program. The greater the work experience, work perception, motivation, and work discipline of the CLC managers, the higher the success of program implementation, with a pure contribution of 53.8%. Meanwhile, 46.2% was determined by other variables. The results of the partial regression analysis for each predictor variable are presented in Table 7 and explained in Table 8.

**Table 7.** Coefficients [a] of regression analysis results.

| Model | | Unstandardized Coefficients | | Standardized Coefficients | T | Sig. | Correlations | |
|---|---|---|---|---|---|---|---|---|
| | | B | Std. Error | Beta | | | Zero-Order | Partial |
| 1 | (Constant) | −3.032 | 2.945 | | −1.029 | 0.306 | | |
| | Work Experience | 0.200 | 0.084 | 0.174 | 2.398 | 0.018 | 0.326 | 0.238 |
| | Work Perception | 0.186 | 0.093 | 0.208 | 1.996 | 0.049 | 0.599 | 0.200 |
| | Work Motivation | 0.311 | 0.128 | 0.255 | 2.434 | 0.017 | 0.618 | 0.241 |
| | Work Discipline | 0.295 | 0.092 | 0.305 | 3.221 | 0.002 | 0.633 | 0.312 |

[a] Dependent variable: success of the equivalency program.

**Table 8.** The results of partial analysis of the effect of work experience, work perception, motivation, and work discipline on the success of the equivalency program.

| Independent Variables | Dependent Variables | r Zero-Order | *Beta* | r Partial | t | *p* | Conclusion |
|---|---|---|---|---|---|---|---|
| Work Experience (X1) | | 0.326 | 0.174 | 0.238 | 2.398 | 0.018 (<0.05) | Significant |
| Work Perception (X2) | Success of the Equivalency Program (Y) | 0.599 | 0.208 | 0.200 | 1.996 | 0.049 (<0.05) | Significant |
| Work Motivation (X3) | | 0.623 | 0.255 | 0.241 | 2.434 | 0.016 (<0.05) | Significant |
| Work Discipline (X4) | | 0.633 | 0.300 | 0.305 | 3.221 | 0.002 (<0.05) | Significant |

The results summarized in Table 7 are interpreted and described in more detail in Table 8.

As shown in Table 8, the results of the partial regression analysis of the manager's work experience variable (X1) on the success of the equivalency program (Y) obtained a zero-order r coefficient of 0.326, a partial r of 0.238, and a *beta* of 0.174, with a t-value of 2398, and error probability (*p*) of 0.018. The obtained t-value of the analysis was greater than the critical t-value (1.658), and the *p*-value was below 0.05. Thus, the null hypothesis (H0) was rejected, and the alternative hypothesis (H1) was accepted. The manager's work experience had a significantly positive effect on the success of the CLC equivalency program. The greater the work experience of the manager, the higher the success of the CLC equivalency program, with a pure correlation of 0.238. The effective contribution was $0.174 \times 0.326 \times 100\% = 5.67\%$, and the relative contribution was $5.67 \div 53.8 = 10.54\%$.

The results of the second partial analysis of the effect of the manager's work perception variable (X2) on the success of the equivalency program (Y) obtained a zero-order r coefficient of 0.599, partial r of 0.200, and a *beta* of 0.208, with a t-value of 1.996 and error probability (*p*) of 0.049. The obtained t-value of the analysis was greater than the critical t-value (1.658), and the *p*-value was below 0.05. Thus, the null hypothesis (H0) was rejected, and the alternative hypothesis (H1) was accepted. The manager's work perception had a significant positive effect on the success of the CLC equivalency program. The better the perception of the manager, the higher the success of the CLC equivalency program, with a pure correlation of 0.200. The effective contribution was $0.208 \times 0.599 \times 100\% = 12.46\%$, and the relative contribution was $12.46 \div 53.8 = 23.2\%$.

The results of the third partial analysis of the effect of the manager's work motivation variable (X3) on the success of the equivalency program (Y) obtained a zero-order r coefficient of 0.623, a partial r of 0.241, and a *beta* of 0.255, with a t-value of 2.434 and an error probability (*p*) of 0.016. The obtained t-value of the analysis was greater than the critical t-value (1.658), and the *p*-value was below 0.05. Thus, the null hypothesis (H0) was rejected, and the alternative hypothesis (H1) was accepted. The manager's work motivation had a significant positive effect on the success of the CLC equivalency program. The more motivated the manager, the higher the success of the CLC equivalency program, with a pure correlation of 0.241. The effective contribution was $0.255 \times 0.623 \times 100\% = 15.89\%$, and the relative contribution was $15.89 \div 53.8 = 29.53\%$.

The results of the fourth partial analysis of the influence of the manager's work discipline variable (X4) on the success of the equivalency program (Y) obtained a zero-order r coefficient of 0.633, partial r of 0.305, and a *beta* of 0.300, with a t-value of 3.221, and error probability (*p*) of 0.002. The t-value obtained from the analysis was greater than the critical t-value (1.658), and the *p*-value was below 0.05. Thus, the null hypothesis (H0) was rejected, and the alternative hypothesis (H1) was accepted. Work discipline had a significant positive effect on the success of the CLC equivalency program. The more disciplined the manager, the higher the success of the CLC equivalency program, with a

pure correlation of 0.305. The effective contribution was $0.300 \times 0.633 \times 100\% = 18.99\%$, and the relative contribution was $18.99 \div 53.8 = 35.3\%$.

Based on the results of the simultaneous and partial analyses, it could be concluded that the work experience, work perception, motivation, and work discipline of the managers are dominant variables that affect the success of the CLC equivalency program. In addition, the four independent variables had a strong predictive power on the dependent variable. The success of the equivalency program (Y) was determined by the work experience (X1), work perception (X2), work motivation (X3), and work discipline (X4) of the managers implementing the equivalency education programs in the CLC, with the regression equation: $Y = -3.032 + 0.200 X1 + 0.186 X2 + 0.311 X3 + 0.295 X4$.

## 4. Discussion

The findings of the first analysis indicated that work experience affected the success of the CLC equivalency program. This finding is in line with the empirical research results and theory, which state that an increase in one's work experience is followed by a dynamic and progressive increase in learning experience related to the work one is engaged in. Furthermore, cognition and skills related to delegated tasks will continue to increase and strengthen over time. The longer an individual works, the more successful he or she will be in carrying out the task [12,17]. The results of research by Rasyad et al. (2019) showed that the quality of instructors was a dominant variable in the success of the training program [2]. The experience of the instructor has a dominant role in improving the learning outcomes of the trainees. The results of the research by Wiyono and Triwiyanto (2018) showed that the degree of activity in teacher working group meetings had a significant effect on teacher competence [30]. Teacher working group meetings were managed by experienced teachers. Samawi et al.'s (2019) research showed that an effective supervision program based on an organizational culture is greatly dependent on the work experience of the principal [31]. Moreover, Wiyono et al. (2017) showed that eight effective supervision techniques required the role of experienced supervisors for improving teacher performance [32]. The results of the study by Wiyono et al. also showed that the quality of the principal's management had a dominant effect on the quality of school output [32,33]. The quality of a principal's management style greatly depends on the experience of the principal. Thus, the findings of this study are in line with the results of previous studies, which showed that the experience of the leadership had a significant effect on the success of the education program.

The findings of the second analysis indicate that the manager's job perception influenced the success of the equivalency program. This is in line with the theory that states that individual aspects, especially job perception, are very important to identify in detail for use as assessment criteria for each leader involved in decision making and various organizational interests. The job perception variable has a series of contributions to various manifestations of behaviors displayed by staff, regardless of their positions [16]. This finding is also in line with the results of previous studies. Another study by Alwi et al. (2020) showed that personality and attitudes affected the organizational citizenship behavior of teachers [24]. Perception is a component of the attitude towards the organization. The results of the research by Citriadin et al. (2019) also showed that attitudes towards the teaching profession affected teacher performance [22]. The results of the research by Setiyowati et al. (2019) indicated that basic attitudes affected counseling competence. The results of these studies indicate that attitudes affect performance [34]. A good perception of work will lead to a positive attitude. In turn, a positive attitude will affect performance. As a result, good performance will contribute to the success of the program.

The third research finding showed that work motivation affected the success of the equivalency program. The findings of this study confirmed the results of previous studies on the effect of work motivation on staff performance in various business organizations. Motivation was previously reported to have an effect of 0.477 on commitment, and the path from motivation to performance through commitment was 0.758 [19]. Several other studies showed that work motivation had a strong influence on organizational performance



and effectiveness [35]. Thus, the findings of this study support the results of previous studies [36–38].

The fourth research finding showed that work discipline affected the success of the equivalency program. This finding is supported by the results of previous studies, which concluded that the work discipline of staff affected the performance and success of the company [18]. The results of Sofian et al.'s (2019) research showed that discipline affected work performance [21]. Amalia et al.'s (2019) research indicated that the factors that shaped a teacher's discipline included a commitment to goals, the work environment, and leadership, which can be set as an example [39]. Increasing these factors can raise the discipline of work personnel, and, as the discipline of the personnel increases, their work may also improve the success of the program. The correlation coefficient and predictions showed that, among the analyzed variables, work discipline had the highest contribution to the success of the equivalency program. This is because work discipline is reflected in behavior, so it has the highest impact on the results achieved. Thus, the findings of this study confirmed the results of previous studies.

The findings of the fifth analysis indicated that the four independent variables, namely, work experience, work perception, work motivation, and work discipline, simultaneously affected the success of the CLC equivalency program. The variable with the greatest contribution to program success was work discipline, followed successively by work motivation, perception of work, and work experience. This is in line with the results of previous studies that reported that work discipline and work motivation had a very strong influence on performance [35,40]. Good performance contributes to the quality or success of the program [32,41]. Work discipline was the dominant factor that supports performance. In addition, work discipline is an aspect that is very closely related to work motivation. High work motivation tends to be followed by high work discipline. Work motivation is manifested in work discipline. Therefore, the effect on performance is also high. A similar pattern applies to work experience: someone who has a lot of experience will find it easier to manage the organization, which will have an impact on success in managing. In addition, work perception had a significant effect on the success of the program. This is in line with the theory that perception is within the individual. Starting with a positive perception will lead to good behavior, and good behavior will be able to translate to the success of program implementation [42,43]. Therefore, this explains why this variable has a weaker effect than the other variables. However, it is important to underline that to maintain or increase the success of the program, the four variables need to be considered because they significantly contribute to the success of the program.

Based on the findings of this study, it can be concluded that psychological factors generally had a strong influence on the success of managers. The results of the study corroborate the results of previous research, which suggested that aspects of individual psychology had a strong influence on a person's success. Individuals who have a strong psychological character will reflect this strength in their behavior [21,22,44]. Strong behavior will lead to success. These psychological aspects include perceptions, motivations, interests, attitudes, talents, experiences, morale, and other psychological attributes. Moreover, in managing the organization, these psychological attributes have a dominant influence on the success of the managers. Some local research results showed that the characteristics of managers or leaders affect the organization's performance [24,42]. Other research from several countries also indicated the same results [45–49]. Many previous research results showed that the psychological attributes of leaders or managers greatly determined their success in managing institutions or organizations.

## 5. Conclusions

Based on the data analysis, several conclusions can be drawn from this study. First, the success of the CLC equivalency programs, particularly the study groups of packages B and C, equivalent to junior high school and senior high school, respectively, in Malang Regency and Malang City, East Java, Indonesia, was classified as very high. The pass rate for the

equivalency exams of packages B and C reached 96.4% and 98.8%, respectively, with an average of 97.3%. Public support for the implementation of the CLC equivalency program was generally high. The members of the community involved in the implementation of the CLC equivalency program reached an average of 68.6%. Public funds that could be collected to support the implementation of the equivalency program reached 75.5% of the total needs.

The four independent variables of work experience, work perception, work motivation, and work discipline, significantly influenced the success of the CLC equivalency program, both simultaneously and partially. Overall, these four variables contributed to 53.8% of the program success. The partial contribution of each variable differed: the highest was work discipline, followed by work motivation, work perception, and work experience.

Based on the research findings, it can be concluded that the success of the CLC equivalency program greatly depends on the manager. If the performance of the manager is good, then the success of the equivalency program will increase. The performance of the manager can be driven by internal and external factors. The internal factors include motivation, work discipline, and work perception, while external factors are indicated to have a significant effect on work experience.

Based on the findings of this study, several recommendations can be put forward. First, to maintain and increase the success of the CLC equivalency program, CLC managers need to have high discipline, high motivation, good work perception, and adequate work experience. Therefore, it is necessary to consider these variables as the basis of assessments during the admission and recruitment of CLC leaders. Second, the managers of CLCs implementing the existing equivalency program need to constantly improve their work perception, motivation, discipline, and work experience. Managers of the equivalency programs who are lacking in any of these aspects should improve, and those who are excellent should work to maintain or continue improving them. Third, to ensure the sustainability of CLCs and community service programs in the fields of education and empowerment, community support for the implementation of the equivalency programs should be maintained and developed. For this reason, the leaders of society should play a role in increasing community support for the implementation of such programs.

The findings of this study are not decisive. Therefore, it is necessary to conduct further research with a wider scope. In terms of substance, several other individual attributes, both permanent (static) and dynamic, need to be studied in greater depth, and other external variables need to be studied further. In terms of methodology, the results would be more robust if studied from a wider perspective by mixing quantitative and qualitative approaches to obtain more comprehensive and generalizable findings.

**Author Contributions:** Conceptualization, S.; Methodology, S.; Software, S. and A.R.; Validation, S. and B.B.W.; Formal Analysis, B.B.W.; Investigation, S. and B.B.W.; Resources, S. and U.D.; Data Curation, S. and L.P.; writing—original draft preparation, S.; writing—review and editing, S. and B.B.W.; visualition, S. and U.D.; Supervision, A.R., B.B.W. and U.D.; Project Administration, S., U.D., L.P. All authors have read and agreed to the published version of the manuscript.

**Funding:** This research was funded by Universitas Negeri Malang and authors.

**Institutional Review Board Statement:** Not applicable.

**Informed Consent Statement:** This study uses a questionnaire (summated rating) for the main data collection and documentation, so that by filling out or answering the items of the research instrument, basically the respondents have agreed to the research.

**Data Availability Statement:** Not applicable.

**Acknowledgments:** The authors would like to acknowledge the support of the Faculty of Education Universitas Negeri Malang Indonesia.

**Conflicts of Interest:** The authors declare no conflict of interest.

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
