# Peer review of "The Contribution of Individual Characteristics of Managers to the Success of Equivalency Education Programs of the Community Learning Center in Indonesia"

_sustainability, doi:10.3390/su131911001_

Round 1

Reviewer 1 Report

This ms. is lacking vital information.

E.g, with regard to the CLC. What are the precise contents, who are the participants, how many participants are there, why do they participate, is it fulltime or parttime, does it cover all of the subjects or only some of them, who are the teachers, how are the CLCs structured and organized, etc.?

When did the study take place?

Present features of the managers. What are their (educational) backgrounds and experience? 

Present the full questionnaire. And describe the results.

What other characteristics of the managers were collected? How? Present the results.

What were the characteristics of the students? Did the authors correct for these differences in the regression analyses?

The analyses presented are very basic, simple. Why not include more relevant variables?

Etc., etc.

Author Response

A Manuscript entitled “The Contribution of Individual Characteristics of Managers on the Success of  Equality Education Programs of  the Community Learning Center in Indonesia”.

Point by Point Response to Reviewers’ Comments

Reviewer 1

Comments

E.g, with regard to the CLC. What are the precise contents, who are the participants, how many participants are there, why do they participate, is it fulltime or parttime, does it cover all of the subjects or only some of them, who are the teachers, how are the CLCs structured and organized, etc.?

When did the study take place?

Present features of the managers. What are their (educational) backgrounds and experience?

Present the full questionnaire. And describe the results.

What other characteristics of the managers were collected? How? Present the results.

What were the characteristics of the students? Did the authors correct for these differences in the regression analyses?

The analyses presented are very basic, simple. Why not include more relevant variables?

Etc., etc.

Responses

  1. Based on Reviewer 1’s comments, the participant characteristics are described clearly, including the characteristic of managers, students, teachers, and staff, namely on pages 7-8. In addition, the organization and program characteritics is explained as well, namely on pages 8-9.
  2. The time for conducting the research is also explained, namely on page 6.
  3. The research instruments are described more clearly, including the dimensions measured, the items of the instrument, and validity, and the reliablity, which is on page 9.
  4. The research sample was added and reanalyzed with descriptive analysis techniques and regression analysis in more detail, and the results are presented in more detail, namely on pages 10-13. The results of the manager's characteristic analysis are also presented on pages 8 and 10.

Reviewer 2 Report

The work is interesting and relevant to sustainable education.  However, it would be convenient to develop the introduction with a more consolidated theoretical framework, linking it with significant contributions at the global level.  I miss references to relevant authors in this regard.
Regarding the research design.  In education I do not believe that moving by cause and effect is the most convenient, but that correlations and descriptions of facts constitute the real richness.

Author Response

A Manuscript entitled “The Contribution of Individual Characteristics of Managers on the Success of  Equality Education Programs of  the Community Learning Center in Indonesia”.

Point by Point Response to Reviewers’ Comments

Reviewer 2

Comments

The work is interesting and relevant to sustainable education.  However, it would be convenient to develop the introduction with a more consolidated theoretical framework, linking it with significant contributions at the global level.  I miss references to relevant authors in this regard.

Regarding the research design.  In education I do not believe that moving by cause and effect is the most convenient, but that correlations and descriptions of facts constitute the real richness.

Responses

  1. The theoretical framework is presented more systematically, and the formulation of research objectives and hypotheses is added, namely on pages 2 and 6. The research methodology is revised and described more clearly, namely on pages 7-9.
  2. The research results are described in more detail, namely on pages 10-13.
  3. The discussion of research results is added, and references are also added, namely on pages 14, 15, and 19.

Reviewer 3 Report

The article is undoubtedly an interesting contribution to comparative considerations. On the other hand, it is addressed to a relatively small group of specialized readers (researchers). It results from the fact that it concerns the cultural, social and educational reality of Indonesia, and therefore embedded in a reality that is little known even to experienced researchers from outside Asia. Hence, certain aspects of the phenomenon under discussion may undoubtedly escape in the face of ignorance of the specificity of this country and its society. Which the authors of the study probably know very well. However, it cannot be expected that these important threads could be included in the article. The authors conclude that this is a presentation of some of the research results that will be continued. Perhaps - this is my suggestion - they will become part of a full monograph. And it will contain threads introducing into the "climate" of Indonesian cultural, social and educational reality. Thus, the circle of beneficiaries of the study may expand - researchers from other regions of the world. The presented text, in my opinion (with the above notes), is technically and methodologically correct. I rate it as inspiring with cognitive values, although, as I wrote above, not fully transparent to the reader, like a reviewer, from Europe. Due to the above-mentioned doubts, my assessment is somewhat "mixed", with an indication of the average, which, however, does not discredit a reliable and conscientious study.

Author Response

A Manuscript entitled “The Contribution of Individual Characteristics of Managers on the Success of  Equality Education Programs of  the Community Learning Center in Indonesia”.

Point by Point Response to Reviewers’ Comments

Reviewer 3

Comments

The article is undoubtedly an interesting contribution to comparative considerations. On the other hand, it is addressed to a relatively small group of specialized readers (researchers). It results from the fact that it concerns the cultural, social and educational reality of Indonesia, and therefore embedded in a reality that is little known even to experienced researchers from outside Asia. Hence, certain aspects of the phenomenon under discussion may undoubtedly escape in the face of ignorance of the specificity of this country and its society. Which the authors of the study probably know very well. However, it cannot be expected that these important threads could be included in the article. The authors conclude that this is a presentation of some of the research results that will be continued. Perhaps - this is my suggestion - they will become part of a full monograph. And it will contain threads introducing into the "climate" of Indonesian cultural, social and educational reality. Thus, the circle of beneficiaries of the study may expand - researchers from other regions of the world. The presented text, in my opinion (with the above notes), is technically and methodologically correct. I rate it as inspiring with cognitive values, although, as I wrote above, not fully transparent to the reader, like a reviewer, from Europe. Due to the above-mentioned doubts, my assessment is somewhat "mixed", with an indication of the average, which, however, does not discredit a reliable and conscientious study.

Responses

  1. Characteristics location and research subjects are described more clearly, so that they can describe the cultural aspects of the research participants, namely on pages 7-9.
  2. The theory underlying the research is presented more systematically, concerning the formulation of hypotheses, research methodology, research findings, and discussion of research results, so that the implications of findings for the theory development can be seen more clearly, namely on pages 2-6, 7-9, 10-13, and 14-15.
  3. The conclusions and suggestions for further research are clarified, namely on pages 15 -16.

Round 2

Reviewer 1 Report

Much better!

Author Response

Point by Point Response to Reviewers’ Comments

Reviewer 1 Round 2

Comments

  1. Much better

  2. English language and style are fine/minor spell check required

Responses

  1. The manuscript was thoroughly edited using the MPDI Editing Service.
